# Relaxation of Some Confusions about Confounders

**DOI:** 10.3390/e23111450

**Published:** 2021-10-31

**Authors:** Ádám Zlatniczki, Marcell Stippinger, Zsigmond Benkő, Zoltán Somogyvári, András Telcs

**Affiliations:** 1Department of Computer Science and Information Theory, Budapest University of Technology and Economics, H-1111 Budapest, Hungary; adam.zlatniczki@ericsson.com; 2Ericsson Hungary, H-1117 Budapest, Hungary; 3Wigner Research Centre for Physics, H-1121 Budapest, Hungary; stippinger.marcell@wigner.hu (M.S.); benko.zsigmond@wigner.hu (Z.B.); somogyvari.zoltan@wigner.hu (Z.S.); 4Department of Quantitative Methods, University of Pannonia, H-8200 Veszprém, Hungary

**Keywords:** causality, common cause, geodesic

## Abstract

This work is about observational causal discovery for deterministic and stochastic dynamic systems. We explore what additional knowledge can be gained by the usage of standard conditional independence tests and if the interacting systems are located in a geodesic space.

## 1. Introduction

It is not necessary to emphasize the importance of the concept of causality in science and in the natural sciences in particular. The concept traverses all disciplines, and it is a matter of extensive research fueled by the exponentially increasing available scientific data and computation power. Revealing causal relations between systems via the time series produced by them is one of the most attractive challenges. The first major advancement was due to Granger who used an auto-regressive framework for a practical implementation of the predictive causality principle by Wiener [1].

The very popular Granger [2] method has some theoretical and practical limitations. It is not able to detect hidden common cause and, instead, indicates false directional causal relation between the observed systems (for details of all the pros and cons cf. [3]). Several further methods appeared in the last two decades (for a concise review see Runge [4] or [5,6]). One of the most prominent is the convergent cross mapping method developed by Sugihara [7] to investigate deterministic dynamic systems, which essentially utilizes Takens’ embedding theorem [8]. Stark [9,10] generalized Takens’ result and showed the theoretical limitations to use it for stochastic dynamic systems. For deterministic dynamics, a new approach was presented in a recent work [11] that was based on the comparison of the dimension of the attractors of the given systems and their joint observation.

The present paper investigates the causal relation of a pair of dynamic systems (which might be deterministic or stochastic). Facts are revealed that, to our best knowledge, avoided the attention of previous studies. We show that the common driver is an i.i.d. sequence, shared observational noise, if there is dependence between the systems with the smallest but positive time difference. We also show that, if the pair is located in a non-abstract physical space where the speed of information transfer is known, then direct causation and common cause cases can be distinguished, which, in general, is theoretically impossible.

### Basic Definitions

First, we provide the framework of our investigation. Our aim is to find the causal relationship between two stochastic dynamic systems *X* and *Y* from which we observe the time series ${\left\{x_{i}\right\}}_{i=1}^{n},{\left\{y_{i}\right\}}_{i=1}^{n}$.

**Assumption** **1.**
*We assume that there is a set of systems $S=\left\{X,Y,L\right\},X\in R^{d_{X}}$ and $Y\in R^{d_{Y}},L\in R^{d_{L}},$$D=d_{X}+d_{Y}+d_{L}$, and an external source of noise W such that the process $m_{i}=\left(s_{i},\omega_{i}\right)=\left(\left(x_{i},y_{i},l_{i}^{1},\ldots l_{i}^{m}\right),\omega_{i}\right)\in R^{2D},s_{i}\in R^{D},\omega_{i}\in$$R^{D}$ has a joint distribution. The series $l_{n}\;$are unobserved, hidden series that are not i.i.d. (see Figure 1).*


In what follows, for $\omega_{1}$, we will use ξ emphasizing that it influences *x* and similarly η for $\omega_{2}$ for *y*.

For brevity, we will use the multi-index of involved dimensions: $\underline{d}$$=\left(d_{X},d_{Y},d_{L}\right)$.

**Assumption** **2.**
*The external noise $\omega_{i}\in R^{D}$ is modeled with an unobserved i.i.d. sequence and affects all the systems with independent $\xi ,\eta ,\omega_{l_{i}}$ components. Furthermore, $\omega_{i+1}$ is independent from $S_{0}^{i}={\left\{s_{j}\right\}}_{j=0}^{i}$.*


**Assumption** **3.**
*The process $m_{i}$ is stationary.*


**Assumption** **4.**
*The causal structure of the time series is time invariant and non random.*


In what follows, we use the expression “drives” for all the terms “causes”, “influences” and “injects information” in relation to dynamic systems.

Following [12], we use the next model. The visible and invisible system can be described by a *p*-order Structural Vector Auto-regressive SVAR model:$$m_{n+1}=f\left(m_{n},\ldots ,m_{n-p+1},\omega_{n+1}\right)$$n∈N, and $m_{0}$ follows the stationary distribution of the system. (It is a $SVAR(\underline{d},p)$ process, were $\underline{d}$ is the multi index of dimensions of the variables, and *p* is the order of auto-regression).

The recursion clearly can be transformed with time delay embedding into higher dimension first order SVAR in particular to SVAR(2D×p,1) in short SVAR(1) with
(1)$$M_{n}=\left(m_{n},m_{n-1},\ldots ,m_{n-p+1}\right):$$
(2)$$M_{n+1}=g\left(M_{n}\;,\omega_{n+1}\right)$$We make the same restriction as in [12] that (2) must be recursive in the variables, which ensures that there is no directed functional cycle. Variables with capital letters denote the same “embedding” as in (1).

**Assumption** **5.**
*The process M is exact (cf. Definition 4.3.2. [13]).*


Exactness means that, if the process started from a set with positive probability, then, after a long time, the set in which it can be found has probability one. It is natural to assume exactness given that we work with an observation, and the support of the observed process for us will be the whole set where the process can run and, consequently, has probability one. On the other hand, exactness implies mixing for stationary processes, and, at the same time, a mixing stationary process is α-mixing (or strong mixing). Let us note here that, from strong mixing, ergodicity also follows, but we do not need that fact (see also [13]).

In our discovery scheme, we may allow instantaneous causation between all variables; however, we do not elaborate on that case here. For brevity, that is not reflected in (2). We note that a system like (2) with contemporaneous interaction but without a directed cycle can be rewritten into the form of (2) using time shifts thanks to the acyclic recursivity.

**Definition** **1.**
*We will say that X drives Y if there is a k>0 s.t.*

(3)
$$Y_{n+k}=f\left(X_{n},Y_{n},L_{n},\eta_{n+k}\right)$$

*where $L_{n}$ stands for the set of latent variables, $\eta_{n+1}$ is an i.i.d sequence that is independent of ${\left(X_{i},Y_{i},L_{i}\right)}_{i=0}^{n}$, and $X_{n}$ cannot be omitted without violating the validity of (3).*


Let us explain that key definition. We may say that there is no such a function *g* that
(4)$$Y_{n+k}=g\left(Y_{n},L_{n},\eta_{n+k}\right)$$
which makes the fact explicit, that $Y_{n+1}$ can be created without $X_{n}.$ Here, one should also observe that the i.i.d. part is also the same as in (3), and there is no possibility for an i.i.d. $X_{n}$ to be hidden in $\eta_{n+1}$.

## 2. Causal Discovery Schemes

The literature of causal discovery is huge. This work has been inspired by two recent ones with their strengths and limitations. First, we found the framework defined by Malinsky in [12] very appealing and that the complex nature of assumptions and the suggested algorithm in [14] presented an essential challenge. The algorithm in [12] is an extension of [15,16]. The algorithm provides a theoretically complete recall of the underlying causal structure at the price that some relations are marked undetermined and some causal relations are not or only partially revealed.

In [14], in addition to many other assumptions, it is assumed that all hidden processes that influence an observed one have no memory (Assumption A9 in [14]). That assumption and A6 in [14] cannot be checked. In [12], such restrictions are eliminated. That paper and most of the works based on Pearl’s DAG analysis have theoretical limitations as admitted in [12]. In what follows, we investigate some situations in which that limitation can be relaxed.

Information from *X* to *Y* can be transferred along a chain of direct causal links, along a directed path $\pi_{X,Y}$. The length of the path (the number of intermediate components plus one) is denoted by $l=l{\;}_{X,Y}=l\left(\pi_{X,Y}\right).$ Such a path has a starting and ending time n,n+l (for arbitrary n≥0,l>0), the difference is the time lag.

**Assumption** **6.**
*We assume that, with some background information, the minimal lag between the systems X and Y can be determined.*


We consider the smallest lag τ for which dependence can be detected in “direction” *X* to *Y*:(5)$$\tau =\tau_{X,Y}=\min_{\pi }\left\{l\left(\pi_{X,Y}\right)>0\right\}$$

### 2.1. The Decomposable Case

We introduce our notation. In order to save space, let $\left(A,B\right)=\left(X,Y\right)$ or (Y,X). Let *I* stand for the Shannon entropy/differential entropy based mutual information. We define conditional mutual information between elements of time series $a_{n},b_{n}$ and similarly for other series. A segment from *k* to *l* of a time series $a_{n}$ are denoted by $A_{k}^{l}$. Such segments are used in the condition representing a part or the full past. In order to investigate if there is information transfer from *B* to *A* with a given time lag $\tau_{b,a}$ we use the conditional mutual information between $a_{n+\tau_{b,a}}$ and $b_{n}$ given the full past of both series ${A_{0}}^{n+\tau_{b,a}-1}$ and $B_{0}^{n-1}$, and we denote it by $I_{B}$. We define the following conditional mutual information
$$\begin{array}{ccc} I_{B} & = & I\left(a_{n+\tau_{b,a}};b_{n}|A_{0}^{n+\tau_{b,a1}-1},B_{0}^{n-1}\right),\\ I_{A,B}^{\left(k\right)} & = & I\left(a_{n};b_{n+k}|A_{0\;}^{n-1},B_{0}^{n+k-1}\right),\mathrm{for}\;\mathrm{any}\;0\leq k<\tau_{A,B}. \end{array}$$
where we set $A_{p}^{q}=\left(a_{q},\ldots ,a_{p}\right)$ and similarly for *B* and other variables.

**Proposition** **1.***Let for L,A,B ( $\left(A,B\right)=\left(X,Y\right)$ or (Y,X) )*$$\delta =\delta_{L,A,B}=\tau_{L,B}-\tau_{L,A}\geq 0.$$*Under Assumptions 1–6 for*$\delta_{L,A,B}=k,0\leq k<\tau_{A,B}$*, the following implications hold.*$$\begin{array}{ccc} & & \left\{\begin{array}{ccccc} I_{A}\; & I_{B}\; & & A\to B & B\to A\\ =c^{\prime } & =0 & \Leftrightarrow & \exists & ∄\\ =c^{\prime }\; & =c^{\prime \prime }\; & \Leftrightarrow & \exists & \exists \end{array}\right\}\;\times \left\{\begin{array}{ccc} I_{A,B}^{\left(k\right)}\; & & CD\\ =0 & \Leftrightarrow & ∄\\ =c & \Leftrightarrow & \exists \end{array}\right\},\; \end{array}$$*Relation 1. Logical relations between conditional mutual information values and causal relations where*CD*stands for Common Driver and*$c,c^{\prime },c^{\prime \prime }>0$. *In the right part of the table, =0 means that*$I_{A,B}^{\left(k\right)}=0$*holds for all*$0\leq k<\tau_{A,B}$*, while >0 means that there is at least one such k for which*$I_{A,B}^{\left(k\right)}>0$.

The proposition summarizes the possible inferences in a concise way. In Relation 1, the headers contain the list of possible combinations and the possible causal scenarios. We have the direct product of two lists of cases collected in the two tables. The header of tables contains, on the left, the quantities that are decisive and, on the right, the possible causal scenarios. As an example, in the left table, the first row shows that if and only if we have that $I_{B}=0$ but $I_{A}=c^{\prime }>0$ (significantly differ from zero) then *B* does not drive *A* but *A* drives *B*. In the right table, if $I_{A,B}^{\left(k\right)}=0$ holds for all $0\leq k<\tau_{a,b}$ that means that there is no common information between members of the series for $k<\tau_{a,b}$, while, in the opposite case, there should be a common driver, given, that there is shared information that cannot be attributed to driving with a lag below $\tau_{a,b}$. If $\delta =\tau_{x,y}$ (or $=\tau_{y,x}$) then, causation between *X* and *Y* and a common driver may coexist, and we cannot separate those models. In the next section we provide some observations in that situation.

### 2.2. The Confounder Case

We assume that $\;\delta_{L,X,Y}=\tau_{X,Y}$ but τ>0. If τ>0, we can investigate the common information between $X_{n+1}$ and $Y_{n+\tau }$. Unfortunately, the variables $X_{n},Y_{n+\tau }$ have a confounder; therefore, we cannot tell which causal relation is behind the dependence. However, some internal structure can be revealed. In line with the assumptions $\;\delta_{L,X,Y}=\tau_{X,Y}$ but τ>0, we assume that
(6)$$\begin{array}{ccc} I_{0} & = & I\left(X_{n};Y_{n+\tau }|X_{0\;}^{n-1},Y_{0}^{n+\tau -1}\right)>0 \end{array}$$
(7)$$\begin{array}{ccc} I_{1} & = & I\left(X_{n+1};Y_{n+\tau }|X_{0\;}^{n},Y_{0}^{n+\tau -1}\right)=0 \end{array}$$Let $b_{1}$ be the information that is passed from $X_{n}$ to $Y_{n+\tau }$ and $b_{i}$ for i=1,2 from an *L* to both (if one or other information transfer takes place). We also let $a_{1}$ be the information passed from $X_{n}$ to the $X_{n+1}$ as Figure 2 shows.

From (7), we have that $b_{1}$ is independent from $a_{1}$ and $b_{1}$ is independent from $b_{2}$. Thus, we have that the information $b_{n}$ injected to $X_{n}$ and $Y_{n}$ from *L* is an i.i.d. sequence. A similar argument shows that the information $c_{1}$ passed from $Y_{n+\tau }$ to $Y_{n+\tau +1}$ is independent from $b_{2}.$ We still cannot decide if *X* drives *Y* or *L* drives both; however, in the latter situation, we may say that *L* emits observational noise for *X*, and it does not influence its evolution (the value of $a_{i}$). Alternatively, we may consider $b_{i}$ as the “part” of *X*, which is injected to *Y*. Let us note that *L* itself is not necessarily an i.i.d. sequence but, from the point of view of its impact on *X* and *Y*, it is indifferent.

One may appeal to the Occam’s razor principle (if other background knowledge does not dictate otherwise) that *L* itself is an i.i.d. process. If *b* is part of *X* or external noise that cannot be decided without further knowledge, we may refer again to the Occam’s razor principle and assume that there is no a third system, a common driver but *X* injects an i.i.d. sequence to *Y*.

### 2.3. Geodesic Spaces

Now, we investigate the case when the subsystems of *M* are located in a geodesic metric space with unique geodesics between any pair of points. We assume that the information transfer speed is uniform, constant in the space regardless of the location of the source and target. Under that assumption, we can speak interchangeably about distance in space and time.

### 2.4. Strict Reversed Triangular Inequality

If $\delta =\min_{L}\delta_{L,X,Y}$ and
(8)δ>τ
then we have
(9)$$\tau_{L,Y}>\tau_{L,X}+\tau_{X,Y}$$
the reversed, strict triangular inequality, and there is information share between $X_{n}$ and $Y_{n+\tau }$, then no *L* can be a common driver of $X_{n}$ and $Y_{n+\tau }$ (cf. Figure 3), so a direct driving should take place from *X* to Y.

### 2.5. Strict Triangular Inequality

On the other hand, if for an *L*
(10)$$\delta_{L\;X,Y}<\tau_{X,Y}$$
then we have
(11)$$\tau_{L,Y}<\tau_{L,X}+\tau_{X,Y}$$
and $X_{n}$ and $Y_{n+\tau }$ have positive conditional mutual information conditioned on the past, then only L, the common driver can produce it, not causation (see Figure 4).

### 2.6. The Equality, the Confounder Case

Finally, if
$$\tau_{X,Y}=\delta_{L,X,Y},$$
(12)$$\tau_{L,Y}=\tau_{L,X}+\tau_{X,Y}$$
we have a confounder.

If the metric space has a unique geodesic from *L* to *Y*, then *X* should be on that geodesic of *L* and *Y*, and this means that the information from *L* either enters *X* along the path to *Y* or avoids it in in a tricky way by an infinitesimal detour as Figure 5 depicts.

In the former case, we have no confounder but the causal chain L→X→Y. This is a situation that, again, cannot be resolved without additional information about the actual systems under scrutiny. Economists used to call such *L* an instrumental variable.

Now, let us recall that the inequalities (8) and (10) read as
(13)$$\begin{array}{ccc} \tau_{L,Y}\; & > & \tau_{L,X}+\tau_{X,Y}, \end{array}$$
(14)$$\begin{array}{ccc} \tau_{L,Y}\; & < & \tau_{L,X}+\tau_{X,Y}. \end{array}$$The latter one is the strict triangular inequality and the former one is its converse (both with strict inequality). Here, we arrive at the interpretation of causation in *M*. If it is a system in an abstract space without metric properties, there is no point to speak about distances in it, and there is no link between information transfer time (lag in short) and distances.

On the other hand, if
the system *M* is located in a geodesic metric space,the geodesics are unique,the information propagates along the geodesics, andthe information transfer has a constant speed,
then, distances are proportional to the delay with the same constant factor for all members. Triangular inequality is inherited from distances to lags. In the case of a metric space, like the Euclidean, hyperbolic and spherical with unique geodesic (except if *X* and *Y* are the oppositely positioned on the sphere) the triangular inequality holds, and thus (13) is impossible, and *L* cannot be a common driver that mimics driving or acts parallel to a driving between *X* and Y. Let us note that the triangular inequality holds for space-like vectors in the Minkovsky space, while the converse holds for time-like positions. Finally, the case of strict equality needs further investigation.

In case of different transfer speed, the picture is more complex, and the above geometric consideration is applicable in particular settings only. In the human brain, the information transfer has different speed depending on the transfer mode: via sequences of neural cells, long axon bundles or volume of surface currents. The transfer speed depends on the number of intermediate relay nodes of the network as well. Consequently, the case of causality analysis of brain regions needs detailed information on the connection type and speed between them. It is likely that many other topical areas, like climate and geophysics, specific knowledge of the metric properties and transfer speed may contribute to the success of causal discovery. In other areas, there is no information about the temporal arrangement of the unobserved factors, and consequently revealing the perfect description of the causal structure seems impossible.

### 2.7. Conditions and Mixing

Let us recall here that all the methods that are based on Pearl’s DAG analysis use d-separation ( or causal Markovness) based on a conditional independence test (CIT) in which parents are the conditioning variables. As such, they need access to the parents, which is impossible if those are not observed, and the computation cost can be prohibitive for large networks. Let us see that the d-separation uses the parents as cut set in the DAG. In Section 2, we used the full past of both observed processes. In practice, it is impossible to put the whole past in the condition; therefore, we should work with a shorter history. Let us consider, as an example, the case when $0\leq k<\tau_{xy}$, which means that there is no information transfer from $x_{n}$ to $y_{n+k}$ and investigate $I\left(x_{n\;};y_{n+k}|X_{0}^{n-1},Y_{0}^{n+k-1}\right)$. If $I\left(x_{n\;};y_{n+k}|X_{0}^{n-1},Y_{0}^{n+k-1}\right)=0$, i.e., there is no hidden common driver. One can show that
$$I\left(x_{n\;};y_{n+k}|X_{0}^{n-1},Y_{0}^{n+k-1}\right)\;\leq \;I\left(x_{n\;};y_{n+k}|J_{d}\right)\to 0$$
as d→∞, where $J_{d}=\left(X_{n-d}^{n-1},Y_{n\;-d}^{n+k-1}\right)$. For the proof, see Appendix A.

With this argument, we have that the convergence to a constant or to zero of the conditional mutual information determines if there is a driving between *X* and *Y* and if there is a common driver (as indicated in Relation 1). Under Assumption 5, it is evident that if there is a hidden common driver, the information is passed along a fixed length path from the common cause to *X* and *Y*, and its effect on dependency is not diminishing. If there is no common driver, the exchanged information should traverse longer and longer paths, and the Conditional Mutual Information (CMI) should go to zero as *d* goes to infinity.

The conditional independence test (and proper estimate of CMI) has recently been the focus of research motivated by applications in machine learning and artificial intelligence. This is known to be a challenging task (cf. [6,12,17,18,19]).

## 3. Related Works and Discussion

There are numerous extensions and refinements of the original PC algorithm that Pearl developed [20]. This applies to the study of causal discovery of dynamic systems based on observed time series. We mention some prominent works [4,6,12,21,22] and their bibliography for further reading (see also the extended surveys [5,23]). The recent works [12,14] (see also [23]) have a very similar approach to the present one. In particular, we also use the structural modeling framework; however, we limit our focus to the discovery of a causal relation between a pair of systems. The method can be extended to the study of many time series by considering vector valued observations and/or many pairwise investigations.

The capabilities and limitations of the causal discovery algorithms were investigated in detail in seminal works [15,16,20,24] and recently in [14,21]. The recent generalizations are complete. They extend the labeling of edge ends of classical DAGs, while completeness does not mean that all relations are well specified. Completeness means that all the possible MAGs (Markov Equivalent Acyclic Graphs) can be created.

In this paper, we used an essential assumption and two unavoidable approximations. First, we assumed that the continuous time process can be inferred using a discrete time and limited resolution time series observation. Next, we assumed that the discrete time process can be well approximated with an order-p SVAR model. Finally, if the processes contain continuous variables, the condition is not restricted to a single state value but to a set of them, and, as a consequence, it is not perfectly blocking the information flow between the marginal variables. This deficiency might be eliminated by the local permutation method proposed by Runge in [19].

## Figures and Tables

**Figure 1 entropy-23-01450-f001:**
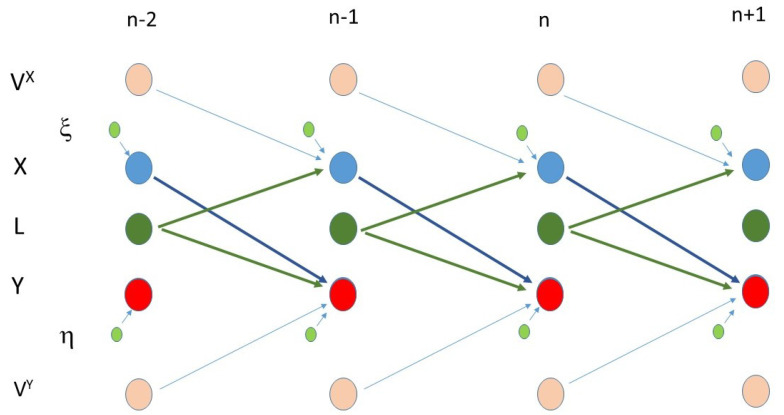
An example is given for a possible causation scheme for the system *M*. In the observed series *X* and *Y*, and in between we have *L* a common cause. Above *X* and below *Y*, small circles represent the i.i.d. input ξ, η, and the large circles $V^{X},V^{Y}$ (also belonging to the set of unobserved series) represent the non i.i.d. influences that are not shared and not common for *X* and *Y*. In this example, *X* drives *Y*, and they have *L* as a common cause.

**Figure 2 entropy-23-01450-f002:**
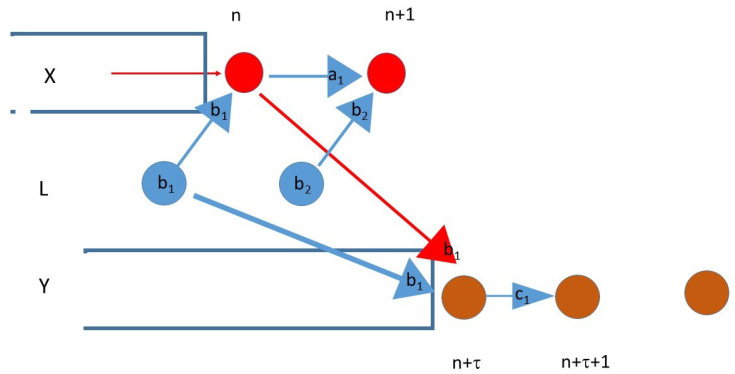
The lag τ and lag difference δ are equal.

**Figure 3 entropy-23-01450-f003:**
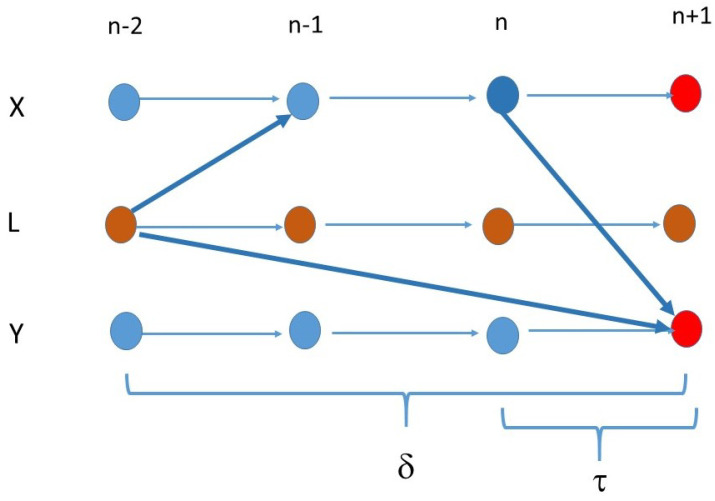
The causation has smaller time lag τ compared with the difference δ from the common driver.

**Figure 4 entropy-23-01450-f004:**
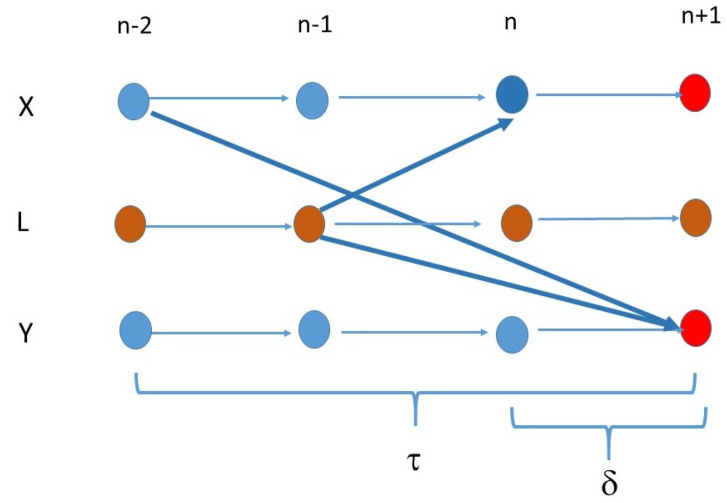
The causation has larger time lag τ compared with the difference δ from the common driver.

**Figure 5 entropy-23-01450-f005:**
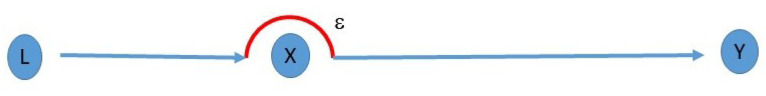
The causation time lag τ equal to the lag difference δ.

## Data Availability

Not applicable.

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
