# Peer review of "Relaxation of Some Confusions about Confounders"

_entropy, 2021, doi:10.3390/e23111450_

Round 1

Reviewer 1 Report

The paper is well-written and quite interesting. Nevertheless, I had some difficulties reading and understanding some of your equations. I suggest that you provide further remarks and explanations regarding some formulae. For example, the functions I_B and I_{A,B) (l. 98). At first view, table 1 is also not obvious to read. Some typos were also detected:

l.73 We may say that...

l.99 where a vector A_p^q... ---> the sentence seems incomplete.

l.106 Rewrite the sentence. 

l.107 significantly different from zero

l.116 We know that ---> from where?

l.156 the inequalities (9 and 11) read as...

l.167 distances are proportional to...

l.168 In the case of metric spaces,...

l.200 as indicated in Figure 6

Author Response

Dear Referee,

The authors express their thanks for the observations.  All the indicated typos and language errors are corrected.  The sentence at l 106 is rephrased.  The notations before Proposition 1, the statement of it and in particular the table in it are explained in details as suggested.

Sincerely yours

András Telcs

Reviewer 2 Report

In lines 219 and 220 the curse of dimensionality is mentioned. I suggest to add some discussion of this problem in relation to the reviewed paper.  

Line 298 is over-fulled. it should be corrected

In formula just after line 299, the inequality sign \leq should be put in the second line

Author Response

Dear Referee,

The authors express their thanks for the observations.   The latex errors are corrected in l 298.  The inequality in l 299 moved to the next line.  The comment on curse of dimensionality removed, we think that is almost a commonsense and not worth to mention.

Sincerely yours

András Telcs
